# Physical activity levels, exercise intrinsic motivation, physical fitness, and their association with adiposity and Oxytocin Receptor (*OXTR*) rs53576 and rs2254298 gene variants

Yee-How Say 🔵*, Kristin-Ann Zhe Mun Leong, Hui Wen Ng, Zi Di Ng, Geetha Letchumanan, Jack Bee Chook

Department of Biomedical Sciences, Sir Jeffrey Cheah Sunway Medical School, Faculty of Medical and Life Sciences, Sunway University, Selangor, Malaysia

* yeehows@sunway.edu.my

## Abstract

Intrinsic motivation predicts higher exercise participation and long-term sustenance. Common variants in the oxytocin receptor gene (*OXTR*) have been associated with socially-related personality traits and behaviours, and obesity pathogenesis. The study aims to investigate the association of physical activity (PA) level, intrinsic motivation, and physical fitness, with adiposity and *OXTR* rs53576 and rs2254298 among a sample of Malaysian urban young adults in Sunway University. A total of 273 participants (M/F = 118/155; aged 21.5 ± 2.9) self-reported their socio-demographics, PA levels via International Physical Activity Questionnaire (IPAQ) Short Form, and intrinsic motivation via Motives for Physical Activities Measure – Revised (MPAM-R). Physical fitness was assessed by three-minute step test, while anthropometric and body composition measurements were taken. Genotyping was performed by allele-specific real-time PCR. Men reported higher PA levels and higher Interest, Competence, and Social scores than women. Interest and Competence scores were significantly positively correlated with vigorous, moderate and total METs, and were also significantly associated with Waist-Height Ratio. Fitness was significantly associated with Waist-Hip Ratio. Physical fitness was significantly positively correlated with vigorous and total METs. *OXTR* rs53576 was significantly associated with Appearance only, but not PA levels, physical fitness, and adiposity. Men were more physically active and intrinsically more motivated to exercise than women. The desire to have fun and engage with challenges when exercising correlates with more frequent exercise, and is a predictor of lower adiposity. *OXTR* rs53576 influences motivation for being physically active in order to become more physically attractive.

**Data availability statement:** All relevant data are within the manuscript.

**Funding:** This research is supported by the Malaysian Ministry of Higher Education Fundamental Grant Research Scheme FRGS/1/2022/STG01/SYUC/02/1. The funder had no role in the design of the study; in the collection, analyses, or interpretation of data; in the writing of the manuscript; or in the decision to publish the results.

**Competing interests:** The authors have declared that no competing interests exist.

## Introduction

Physical activity (PA) is a fundamental contributor to physical and mental health, yet participation levels remain suboptimal worldwide, especially during and after the COVID-19 pandemic [1]. In Malaysia, the 2023 National Health and Morbidity Survey (NHMS) reported that 29.9% of adults were physically inactive [2]. Furthermore, half of the adult population spent more than two hours per day in sedentary behaviour while awake, and 84% did not engage in any form of sport, recreational activity, or active commuting such as walking or cycling [2]. These alarming trends signal an urgent need to better understand the underlying drivers of physical inactivity and to inform more effective intervention strategies, particularly in the Malaysian context.

Motivation has been consistently identified as a key psychological determinant of PA participation [3]. Self-determination theory (SDT) provides a widely accepted framework to examine motivation, distinguishing between intrinsic motivation—engaging in activity for enjoyment, interest, or personal growth—and extrinsic motivation, which is driven by external rewards or social pressures [4]. Intrinsic motivation, in particular, has been associated with sustained PA engagement [5]. Its five core dimensions—enjoyment, competence, appearance, fitness, and social connection—may vary significantly across individuals [6]. Research suggests that gender may influence the type and strength of exercise motivation, with men more often motivated by strength, competition, and recognition, while women tend to be influenced by appearance and social factors [7,8]. However, these findings remain inconsistent and are often based on Western populations, limiting their generalizability to Malaysia's sociocultural setting.

Beyond psychological factors, biological influences may also shape PA motivation and behaviour. The oxytocin receptor (*OXTR*) gene, particularly the polymorphisms rs53576 and rs2254298 [9], has been studied in relation to social behaviour, stress reactivity, eating patterns, and obesity [10–13]. Given oxytocin's role in social bonding and its interaction with the brain's reward system [14], these genetic variants may plausibly affect motivation to engage in PA. However, their relevance to exercise motivation and physical activity remains underexplored, especially in Southeast Asian populations.

Current research on the relationship between PA motivation and activity levels is also limited by its focus on younger populations, particularly children and adolescents. Few studies have investigated these relationships in adults, especially young adults aged 18–40 years, who are in a transitional life phase marked by shifting lifestyle habits and health priorities. Moreover, limited studies have been conducted in Malaysia, where environmental, cultural, and genetic factors may interact uniquely to influence PA behaviour.

To address these gaps, this cross-sectional study was conducted with the following objectives: 1. To determine whether there was a significant gender difference in PA levels and the five exercise intrinsic motivations (enjoyment, competence, appearance, fitness, social); 2. To examine the correlation between PA levels, physical fitness, and intrinsic motivation; 3. To investigate the association of PA levels, physical fitness, and intrinsic motivation with adiposity; and 4. To investigate the possible

roles of two common *OXTR* gene variants—rs53576 and rs2254298—in determining PA levels, physical fitness, intrinsic motivation, and adiposity.

By integrating psychological, physiological, and genetic perspectives, this study aims to provide a more comprehensive understanding of the factors influencing physical activity among Malaysian young adults, thereby supporting the development of more personalized and culturally relevant interventions to promote active lifestyles.

## Methods

### Recruitment of participants and ethical approval

Recruitment of participants was carried out from May to June 2023 by convenience sampling. This was done through the distribution of flyers around Sunway College and Sunway University campus and through social media platforms. The inclusion criteria for this study were that the participants must be: (1) 18 years old and above, (2) a student or staff from Sunway College or Sunway University, (3) fluent in English, (4) able to walk without an assistive device, and (5) have no heart, lung, blood, muscle, or bone problems that prevent them from walking, standing from a chair, or climbing steps. Using the Raosoft sample size online calculator (http://www.raosoft.com/samplesize.html), a minimum sample size of 264 is needed to achieve a 6% margin of error, 90% confidence level, Sunway University and Sunway College population size of 20,000, and a 50% response distribution.

The study was conducted in accordance with the Declaration of Helsinki and approved by the Sunway University Research Ethics Committee (SUREC 2023/012). Online written informed consent was obtained from all subjects involved in the study.

### Questionnaires

First, participants interested in participating were checked to ensure they fulfilled all the inclusion criteria. Next, eligible participants were asked to scan a QR code which led to an online Google Form containing the participant information sheet, consent form, and the questionnaires. Participants were asked to read the participant information sheet and give their online written voluntary consent before they were allowed to proceed to the questionnaires.

**Sociodemographic and lifestyle factors.** The first part of the questionnaire contained questions regarding the sociodemographic and lifestyle factors of participants. Questions regarding the name, age, gender, ethnicity, marital status, highest educational level, monthly household income, smoking habit, and alcohol consumption of participants were included.

**International Physical Activity Questionnaire – Short Form (IPAQ-SF).** The second part of the questionnaire contained the IPAQ-SF [15] which is a self-reported questionnaire used to assess the frequency of doing PAs of different intensity levels for the past seven days. This measure contains seven open-ended questions surrounding the number of days and time spent (in hours and minutes) for doing vigorous PA, moderate PA, walking, and sitting. The sample items are "During the last 7 days, on how many days did you do vigorous physical activities like heavy lifting, digging, aerobics, or fast bicycling?" and "How much time did you usually spend doing vigorous physical activities on one of those days?". This measure is catered for individuals between the age of 15–69 years old. This measure has shown high reliability and validity [16].

The volume of PA for each type of activity can be interpreted by its energy requirement, which is defined in the unit of metabolic equivalents of task (METs). The MET-mins/week value for each type of PA is obtained using the following formulas: Walking=3.3 x (minutes of activity) × (events per week); Moderate PA=4.0 × (minutes of activity) x (events per week); Vigorous PA=8.0 × (minutes of activity) × (events per week). The total PA level score is obtained using the following formula: (Walking MET-mins/week) + (Moderate MET-mins/week) + (Vigorous MET-mins/week). The total PA level (MET-mins/week) is then classified into three classes; low (<600), moderate (600–2999.99), and high (≥3000) PA [15].

**Motives for Physical Activities Measure - Revised (MPAM-R).** The third part of the questionnaire contained the MPAM-R [6], which was used to assess the reason why people engage in physical activities, sports, and exercise. This

measure contains 30 items on the general motives of exercise participation and is divided into five categories; Interest/ Enjoyment (7 items), Competence (7 items), Appearance (6 items), Fitness (5 items), and Social (5 items). The scoring ranges from 1 to 7, where 1 indicates "not at all true for me" and 7 indicates "very true for me". This measure has shown acceptable validity and reliability with α-value ranging from 0.78 to 0.92 [17]. The scores for the five MPAM categories were further classified into low, medium and high based on their 25th and 75th percentile values as follows: Interest –<3.00, 3.00–5.57,>5.57; Competence –<3.00, 3.00–5.43,>5.43; Appearance –<3.17, 3.17–5.83,>5.83; Fitness –<3.70, 3.70–6.10,>6.10; Social –<2.20, 2.20–4.80,>4.80.

## Physical fitness assessment using the three-minute step test

The three-minute step test was performed to assess cardiorespiratory endurance as a fundamental component of physical fitness [18]. Participants were asked to step on and off a 30 cm step, 24 times per minute for 3 minutes. They were aided in this endeavor by a metronome set at 96 beats per minute (bpm) which they were to match with 96 steps (24 ascent-descent cycles) per minute. Immediately after completion of the test, the participants' pulse rate was measured for 1 min using the SB200 Fingertip Pulse Oximeter (Rossmax International Ltd., Taiwan). Physical fitness was determined based on the pulse rate for the particular age group and gender, and were rated as 1. Excellent, 2. Good, 3. Above Average, 4. Average, 5. Below Average, 5. Poor, and 6. Very Poor (higher score denotes poorer physical fitness) [18]. To further minimize the number of categories, Excellent, Good, and Above Average were recategorized as "Good", while Below Average, Poor, and Very Poor were recategorized as "Poor". Participants were also asked to rate their perceived exertion when performing the test using the Borg Rating of Perceived Exertion (RPE) scale [19], ranging from 6 – no exertion at all to 20 – maximal exertion (higher score denotes higher exertion).

## Anthropometric and body composition measurements

Systolic blood pressure (SBP), diastolic blood pressure (DBP), and pulse rate were taken using an automated blood pressure monitor (HEM-7121, Omron, Japan) after the subjects had rested for 5 min. Height was measured using a wall-mounted stadiometer. Waist and hip circumferences were measured using a stretch-resistant tape that provided a constant 100 g tension, at the midpoint between the lower margin of the least palpable rib and the top of the iliac crest and around the widest portion of the buttocks, respectively [20]. The waist-hip ratio (WHR) and waist-to-height ratio (WHtR) were calculated by dividing waist circumference by hip circumference and height, respectively. A bioimpedance body composition scale (Omron HBF-375) was used to determine weight, body mass index (BMI; kg/m$^2$), total body fat (TBF; %), visceral fat level (VFL; %), subcutaneous fat (SF; %), skeletal muscle percentage (SM; %) and resting metabolism rate (RM; kcal). The cutoff points for overweight, obesity, high TBF, high VFL, high SM, high WC, high WHR and high WHtR are ≥ 23 kg/m$^2$ [21]; ≥ 27.5 kg/m$^2$ [21]; 20% (men) or 30% (women) [22]; 10% [22]; 35.8% (men) or 28% (women) [22]; 90 cm (men) or 80 cm (women) [21]; 0.90 (men) or 0.85 (women) [20]; and 0.50 [23], respectively.

## Genotyping of *OXTR* gene variants

Participants were instructed to rinse their mouths vigorously with 10 ml mouthwash (Listerine® Healthy White, Johnson & Johnson, containing ethanol) for 1 min, with their tongues rubbing their inner cheeks and upper plate, and then spit it into a tube. The samples were stored at room 4°C until further processing. Genomic DNA were extracted from mouthwash samples using the GF-1 Tissue DNA Extraction Kit (Vivantis, Malaysia), before proceeding for genotyping of *OXTR* rs53576 and rs2254298 using the FRET (fluorescent resonance energy transfer) chemistry allele-specific real-time PCR-based KASP™ genotyping assay (LGC Biosearch Technologies, UK) according to the manufacturer's protocols. Using the Genetic Association Study Power Calculator (https://csg.sph.umich.edu/abecasis/cats/gas_power_calculator/), the statistical power for rs53576 and rs2254298 SNPs were 80% and 50%, respectively, assuming that overweight is the "disease" or outcome; the case/control sample size of 207 normal and 66 overweight; the significance level of 0.05; the

disease model is additive; the prevalence of overweight among Malaysians is 31.3% [24]; the disease allele frequency is 0.36 and 0.45 for rs53576 and rs2254298, respectively; and the heterozygous genotype relative risk is 1.532 and 1.316, respectively.

## Statistical analysis

Statistical analysis of the data was performed using IBM SPSS Statistics for Windows 26.0 (IBM Corp., Armonk, NY, USA). Descriptive statistics for the categorical variables (demographic characteristics) were presented in terms of frequency and percentage. The conformity of the numerical variables to normal distribution was determined by the Kolmogorov-Smirnov test, where $p > 0.05$ indicates normally-distributed data. Chi-square test was performed to assess the association of socio-demographic, lifestyle factors, body composition, PA level, intrinsic motivation and physical fitness with gender; the association of PA level, exercise intrinsic motivation and physical fitness with anthropometric classes; and the association of *OXTR* rs53576 and rs2254298 genotypes with demographics, PA level, exercise intrinsic motivation, physical fitness and anthropometric and body composition classes. Mann-Whitney *U* test was performed to compare the means of exercise level, intrinsic motivation, and physical fitness between genders. A partial correlation test was performed to assess the correlation of PA level and physical fitness with exercise intrinsic motivation, controlling for age, gender, ethnicity, marital status, highest education level, and monthly household income. In the interpretation of the correlation coefficient, it was determined as a "very weak correlation, if <0.2", a "weak correlation between 0.2 and 0.4", a "moderate correlation between 0.4 and 0.6", a "high correlation between 0.6 and 0.8", and "0.8> very high correlation" [25]. The *p*-value of < 0.05 was considered as statistically significant.

## Results

### Differences in socio-demographic and lifestyle factors, body composition, PA level, intrinsic motivation and physical fitness classes between genders

Out of 300 participants recruited for the study, 273 participants identified as Malaysians, completed the questionnaires in entirety and had all measurements recorded (dropout rate: 9%). The mean age of the overall participants was 21.5±2.9 years (men: 21.5±2.8; women: 21.4±3.0), with; age range: 18–40 years and; men: women ratio 1: 1.08. Table 1 shows that the majority of them were of Chinese ethnicity, students between 18−25 ages, single, currently pursuing a tertiary education level, were from the M40 monthly household income category (considered as middle-income group with the monthly income of MYR 4851−10,960), were not currently smoking, and had monthly or less alcohol drinking. Majority also had moderate PA level, poor physical fitness, but were not obese or had high adiposity (Table 1). The frequency distribution of sociodemographics (i.e., ethnicity, age, marital status, highest education level), current smoking status, physical fitness category, WC, WHR, BMI Obese and SM classes did not differ significantly between genders (Table 1). However, there were significantly more men than women who were from the lower income group, drank alcohol more frequently, exercised more frequently, had higher Competence and Social, and belonged to the high blood pressure, WHtR, TBF, VFL and overweight categories (Table 1).

Indeed, when means between genders were compared, men reported significantly higher vigorous, moderate and total PA levels, and higher MPAM Interest, Competence, and Social scores than women (Fig 1).

### Correlation of PA levels and physical fitness with exercise intrinsic motivation

Table 2 shows the correlation of PA levels (vigorous MET, moderate MET, walking MET, total MET, and weekday sitting times) and physical fitness (TMST fitness score) with exercise intrinsic motivation (Interest, Competence, Appearance, Fitness, Social scores). Vigorous MET was significantly positively correlated with the scores of all five categories of exercise intrinsic motivation, albeit very weakly or weakly (Table 2). Moderate and total METs were significantly positively correlated with Interest and Competence only, while walking MET was significantly negatively correlated with the scores of

**Table 1. Distribution of socio-demographic and lifestyle factors, body composition, total PA level, intrinsic motivation and physical fitness classes between genders.**

| | Men (*n*=118) | Women (*n*=155) | Total (*N*=273) |
|---|---|---|---|
| **Ethnicity** | | | |
| Malay/Bumiputra | 8 (6.8) | 8 (5.2) | 16 (5.9) |
| Chinese | 99 (83.9) | 130 (83.9) | 229 (83.9) |
| Indian | 11 (9.3) | 17 (11.0) | 28 (10.3) |
| $\chi^2$; *p* | 0.476; 0.788 | | |
| **Age Group** | | | |
| 18-25 | 113 (95.8) | 148 (95.5) | 261 (95.6) |
| 26-40 | 5 (4.2) | 7 (4.5) | 12 (4.4) |
| $\chi^2$; *p* | 0.012; 0.911 | | |
| **Marital Status** | | | |
| Single | 115 (97.5) | 152 (98.1) | 267 (97.8) |
| Married | 3 (2.5) | 3 (1.9) | 6 (2.2) |
| $\chi^2$; *p* | 0.115; 0.735 | | |
| **Highest Education Level** | | | |
| Secondary | 12 (10.2) | 7 (4.5) | 19 (7.0) |
| Tertiary | 106 (89.8) | 148 (95.5) | 254 (93.0) |
| $\chi^2$; *p* | 3.307; 0.069 | | |
| **Household Monthly Income Group**§ | | | |
| B40 | 44 (37.3) | 33 (21.3) | 77 (28.2) |
| M40 | 53 (44.9) | 82(52.9) | 135 (49.5) |
| T40 | 21 (17.8) | 40 (25.8) | 61 (22.3) |
| $\chi^2$; *p* | 8.867; 0.012* | | |
| **Current Smoking Status** | | | |
| Never/Quit | 106 (89.8) | 148 (95.5) | 254 (93.0) |
| Current smoker | 12 (10.2) | 7 (4.5) | 19 (7.0) |
| $\chi^2$; *p* | 3.307; 0.069 | | |
| **Current Drinking Status** | | | |
| Never | 38(32.2) | 58 (37.4) | 96 (35.2) |
| Monthly or less | 61 (51.7) | 89 (57.4) | 150 (54.9) |
| 2-4 times a month | 16 (13.6) | 8 (5.2) | 24 (8.8) |
| 2-3 times a week | 3 (2.5) | 0 | 3 (1.1) |
| $\chi^2$; *p* | 10.233; 0.017* | | |
| **Total PA Level** | | | |
| Low | 6 (5.1) | 29 (18.7) | 35 (12.8) |
| Moderate | 43 (36.4) | 73 (47.1) | 116 (42.5) |
| High | 69 (58.5) | 53 (34.2) | 122 (44.7) |
| $\chi^2$; *p* | 20.330; <0.001** | | |
| **MPAM Interest Category** | | | |
| Low | 23 (19.5) | 40 (25.8) | 63 (23.1) |
| Medium | 56 (47.5) | 80 (51.6) | 136 (49.8) |
| High | 39 (33.1) | 35 (22.6) | 74 (27.1) |
| $\chi^2$; *p* | 4.099; 0.129 | | |
| **MPAM Competence Category** | | | |
| Low | 18 (15.3) | 41 (26.5) | 59 (21.6) |
| Medium | 63 (53.4) | 87 (56.1) | 150 (54.9) |

*(Continued)*

| | Men (*n*=118) | Women (*n*=155) | Total (*N*=273) |
|---|---|---|---|
| High | 37 (31.4) | 27 (17.4) | 64 (23.4) |
| $\chi^2$; *p* | 9.529; 0.009** | | |
| **MPAM Appearance Category** | | | |
| Low | 28 (23.7) | 42 (27.1) | 70 (25.6) |
| Medium | 55 (46.6) | 75 (48.4) | 130 (47.6) |
| High | 35 (29.7) | 38 (24.5) | 73 (26.7) |
| $\chi^2$; *p* | 1.004; 0.605 | | |
| **MPAM Fitness Category** | | | |
| Low | 26 (22.0) | 42 (27.1) | 68 (24.9) |
| Medium | 59 (50.0) | 78 (50.3) | 137 (50.2) |
| High | 33 (28.0) | 35 (22.6) | 68 (24.9) |
| $\chi^2$; *p* | 1.471; 0.479 | | |
| **MPAM Social Category** | | | |
| Low | 19 (16.1) | 43 (27.7) | 62 (22.7) |
| Medium | 64 (54.2) | 85 (54.8) | 149 (54.6) |
| High | 35 (29.7) | 27 (17.4) | 62 (22.7) |
| $\chi^2$; *p* | 8.422;0.015* | | |
| **Physical Fitness Category** | | | |
| Good | 14 (11.9) | 15 (9.7) | 29 (10.6) |
| Average | 16 (13.6) | 10 (6.5) | 26 (9.5) |
| Poor | 88 (74.6) | 130 (83.6) | 218 (79.9) |
| $\chi^2$; *p* | 4.58; 0.101 | | |
| **Blood Pressure Category** | | | |
| Normal | 73 (61.9) | 149 (96.1) | 222 (81.3) |
| Prehypertension | 40 (33.9) | 6 (3.9) | 46 (16.8) |
| Stage 1 hypertension | 4 (3.4) | 0 | 4 (1.5) |
| Stage 2 hypertension | 1 (0.8) | 0 | 1 (0.4) |
| $\chi^2$; *p* | 52.091;<0.001** | | |
| **WC Class** | | | |
| Normal | 108 (91.5) | 145 (93.5) | 253 (92.7) |
| High | 10 (8.5) | 10 (6.5) | 20 (7.3) |
| $\chi^2$; *p* | 0.404; 0.525 | | |
| **WHR Class** | | | |
| Normal | 114 (96.6) | 145 (93.5) | 259 (94.9) |
| High | 4 (3.4) | 10 (6.5) | 14 (5.1) |
| $\chi^2$; *p* | 1.291; 0.256 | | |
| **WHtR Class** | | | |
| Normal | 101 (85.6) | 144 (92.9) | 245 (89.7) |
| High | 17 (14.4) | 11 (7.1) | 28 (10.3) |
| $\chi^2$; *p* | 3.889; 0.049* | | |
| **TBF Class** | | | |
| Normal | 67 (56.8) | 116 (74.8) | 183 (67.0) |
| High | 51 (43.2) | 39 (25.2) | 90 (33.0) |
| $\chi^2$; *p* | 9.887; 0.002** | | |
| **VFL Class** | | | |
| Normal | 104 (88.1) | 152 (98.1) | 256 (93.8) |

*(Continued)*

**Table 1.** (Continued)

| | Men (n = 118) | Women (n = 155) | Total (N = 273) |
|---|---|---|---|
| High | 14 (11.9) | 3 (1.9) | 17 (6.2) |
| $\chi^2$; $p$ | 11.311; 0.001** | | |
| **BMI Overweight Class** | | | |
| Normal | 70 (59.3) | 137 (88.4) | 207 (75.8) |
| Overweight | 48 (40.7) | 18 (11.6) | 66 (24.2) |
| $\chi^2$; $p$ | 30.875; <0.001** | | |
| **BMI Obese Class** | | | |
| Normal | 109 (92.4) | 150 (96.8) | 259 (94.9) |
| Obese | 9 (7.6) | 5 (3.2) | 14 (5.1) |
| $\chi^2$; $p$ | 2.668; 0.102 | | |
| **SM Class** | | | |
| Normal | 72 (61.0) | 100 (64.5) | 172 (63.0) |
| High | 46 (39.0) | 55 (35.5) | 101 (37.0) |
| $\chi^2$; $p$ | 0.352; 0.553 | | |

Parentheses indicate percentages out of the same gender.

§The B40, M40 and T20 categories were ≤ MYR 4,850, 4851−10,960, and ≥ 10,961 (approximately ≤USD 1080, 1080–2,450, and ≥2451), respectively [48].

BMI: Body Mass Index; TBF: Total Body Fat; VFL: Visceral Fat Level; SM: Skeletal Muscle Percentage; WC: Waist Circumference; WHR: Waist-Hip Ratio; WHtR: Waist-Height Ratio; MPAM: Motives for Physical Activities Measure.

*$p$-value is significant at the 0.05 level (2-tailed); **$p$-value is significant at the 0.01 level (2-tailed).

all intrinsic motivation categories, except Social (Table 2). Sedentarism as indicated by sitting time weekdays, were significantly negatively correlated with the scores of all intrinsic motivation categories, except Fitness (Table 2). TMST Fitness Score was significantly negatively correlated with Interest, Competence and Fitness intrinsic motivation scores (Table 2). Finally, TMST Fitness Score was significantly positively correlated with vigorous and total METs ($r = -0.314$, $p = <0.001$, and $r = 0.282$, $p = <0.001$, respectively). As expected, TMST Fitness Score was significantly positively correlated with RPE score, albeit weakly ($r = 0.349$, $p < 0.001$), indicating poorer physical fitness led to higher perceived exertion during exercise.

### Association of PA levels, exercise intrinsic motivation and physical fitness with adiposity

Table 3 shows that total PA level was significantly associated with overweight; more overweight individuals had high total PA level. When stratified according to gender, total PA level was significantly associated with VFL among men ($\chi^2 = 6.942$; $p = 0.031$); lower frequency of those who had high PA level were in high VFL class. For exercise intrinsic motivation, Interest was significantly associated with WC, WHtR and obesity; those who had high WC, high WHtR and obese also had low Interest (Table 3). For the rest of the intrinsic motivation, only Competence was significantly associated with WHtR, and Fitness with WHR; those who had high WHtR and WHR also had low Competence and low Fitness, respectively. Lastly, physical fitness was not significantly associated with adiposity, since majority of the participants belonged to the poor physical fitness category, outnumbering the rest (Table 3).

### Association of *OXTR* rs53576 and rs2254298 with demographics, PA level, exercise intrinsic motivation, physical fitness and adiposity

The minor allele frequencies (MAFs) for rs53576 and rs2254298 were 0.36 and 0.45, respectively. The allele frequency for rs53576 did not differ from Hardy-Weinberg equilibrium ($\chi^2 = 2.139$; $p = 0.144$), but not rs2254298

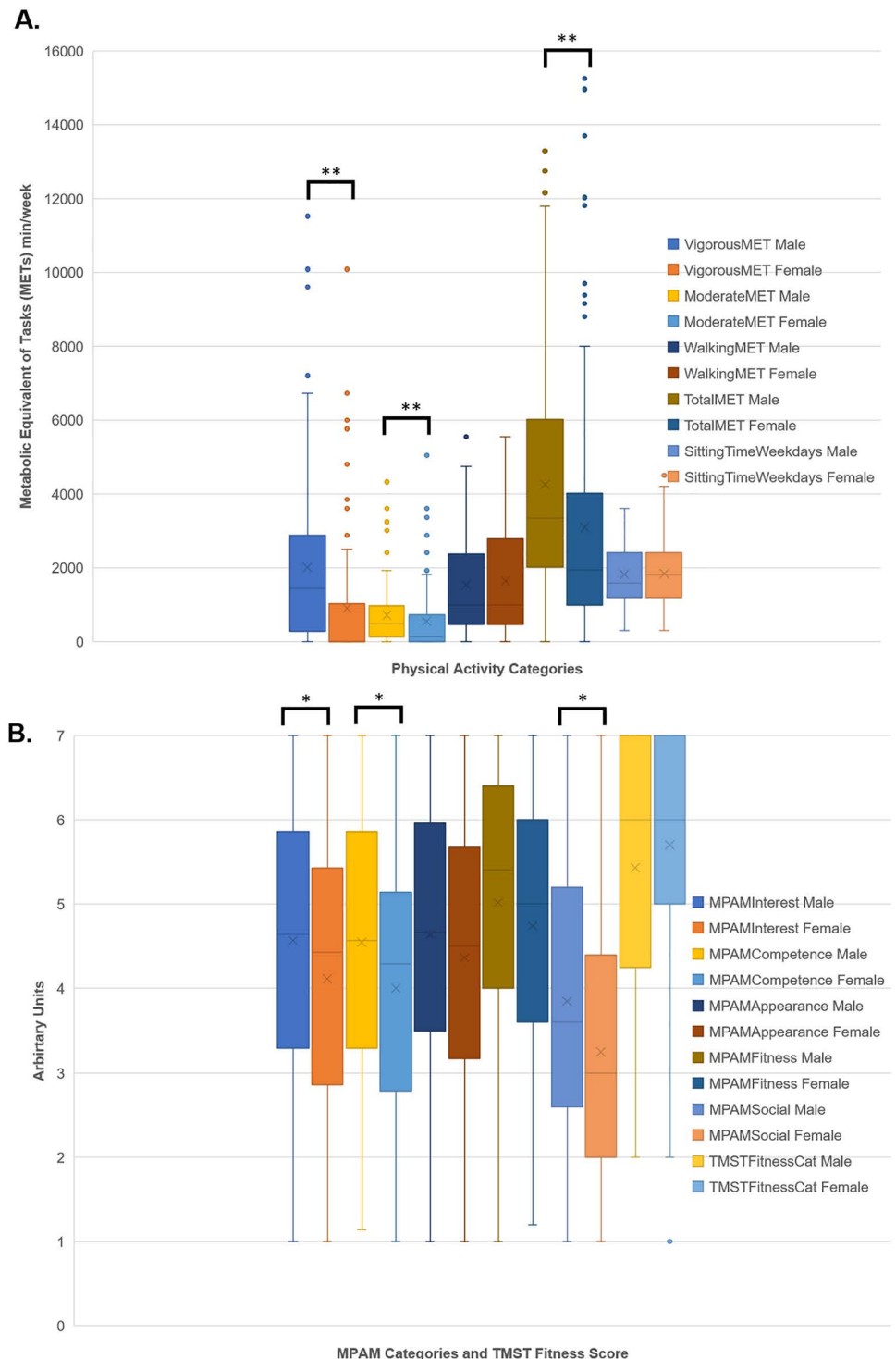

**Fig 1. Comparison of physical activity (PA) levels, physical fitness and exercise intrinsic motivation between genders. A.** Vigorous, moderate, walking, and total Metabolic Equivalent of Task (MET)-min/wk, and weekday sitting times (min); **B.** Scores of the five categories of exercise intrinsic motivation [derived from Motives for Physical Activities Measure (MPAM) - Revised] and three-minute step test (TMST) physical fitness category. The top and bottom sides of the box are the lower and upper quartiles; the box covers the Interquartile Range (IQR); median is represented by the vertical line that split the box in two; cross represents the mean; whiskers at the bottom of box represents lower quartile, whiskers at the top of box represents upper quartile. * indicates p < 0.05, ** indicates p < 0.01; by Mann-Whitney U test.

**Table 2. Correlation of physical activity (PA) levels and physical fitness with intrinsic exercise motivation.**

| PA MET-min/wk/physical fitness score | | MPAM Interest | MPAM Competence | MPAM Appearance | MPAM Fitness | MPAM Social |
|---|---|---|---|---|---|---|
| Vigorous MET-min/wk | r | 0.33 | 0.341 | 0.17 | 0.186 | 0.175 |
| | p | <0.001** | <0.001** | 0.005** | 0.002** | 0.004** |
| Moderate MET-min/wk | r | 0.125 | 0.146 | 0.047 | 0.057 | 0.05 |
| | p | 0.042* | 0.017* | 0.446 | 0.356 | 0.418 |
| Walking MET-min/wk | r | −0.146 | −0.149 | −0.214 | −0.192 | −0.05 |
| | p | 0.017* | 0.015* | <0.001** | 0.002** | 0.417 |
| Total MET-min/wk | r | 0.172 | 0.183 | 0.012 | 0.036 | 0.1 |
| | p | 0.005** | 0.003** | 0.848 | 0.563 | 0.102 |
| Sitting Time Weekdays | r | −0.195 | −0.208 | −0.16 | −0.115 | −0.213 |
| | p | 0.001** | 0.001** | 0.009** | 0.06 | <0.001** |
| TMST Fitness Score | r | −0.162 | −0.177 | −0.063 | −0.123 | −0.014 |
| | p | 0.008** | 0.004** | 0.305 | 0.044* | 0.819 |

Partial correlation test, controlling for age, gender, ethnicity, marital status, highest education level, and monthly household income.

*$p$-value is significant at the 0.05 level (2-tailed); **$p$-value is significant at the 0.01 level (2-tailed).

($\chi^2 = 30.297$; $p < 0.001$). Table 4 shows that there was a significant difference in the frequency distribution of rs53576 genotypes, but not rs2254298, between ethnicities; Indians had higher heterozygous AG and homozygous variant GG genotypes. Similarly, the frequency distribution was significantly different between ethnicities for both rs53576 and rs2254298 alleles ($p < 0.001$; 0.016, respectively; S1 Table). Malays/Bumiputras and Indians had higher frequencies of variant G allele for rs53576, while Indians had a lower frequency of variant A allele for rs2254298; compared with Chinese. Both rs53576 and rs2254298 genotypes were not associated with PA level, exercise intrinsic motivation, physical fitness and adiposity – except for Appearance – where there was higher frequency of those carrying the rs53576 GG genotype who belonged to the category that had high Appearance rating (Table 4). Similarly, rs53576 allele was only significantly associated with Appearance ($p = 0.013$) and Total METs ($p = 0.032$); higher frequencies of those carrying the rs53576 G allele belonged to the category that had high Appearance rating (43.8%) and low PA Category (41.4%) (S1 Table).

## Discussion

This study explored the relationship between PA level, exercise intrinsic motivation, physical fitness, and their association with adiposity and *OXTR* rs53576 and rs2254298 gene variants through a cross-sectional design. The results of this study showed a significant positive correlation between Interest and Competence intrinsic motivations with PA levels (vigorous, moderate, and total), and between physical fitness and PA levels (vigorous and total). *OXTR* gene variants had limited roles, as rs53576 was significantly associated with Appearance intrinsic motivation only, but not PA levels, physical fitness, and adiposity. This study proves that greater exercise motivation and better participation in PA could promote physical fitness. This study is an extension and application of exercise motivation theory based on SDT.

There were significant gender differences in the PA levels and exercise intrinsic motivation scores. Men demonstrated higher levels of vigorous, moderate, and total PA levels and higher MPAM Interest, Competence, and Social scores than women. This finding is consistent with multiple previous studies [26–29]. One of the reasons why women tend to participate less in PA may be because they perceive lesser enjoyment while doing PA [30], while some may feel less competent in their fitness abilities compared to men [31]. Even when they participate, women tend to receive lesser social and parental support or opportunities compared to men in terms of PA [31]. However, Appearance was not significantly higher among women, contrasting with previous studies [27,28], which found that women scored higher for the appearance

**Table 3. Association of physical activity (PA) level, intrinsic exercise motivation and physical fitness with anthropometric classes.**

| | WC Class | | WHR Class | | WHtR Class | | BMI Overweight | | BMI Obese | |
|---|---|---|---|---|---|---|---|---|---|---|
| | Normal | High | Normal | High | Normal | High | Normal | Overweight | Normal | Obese |
| **Total PA Level** | | | | | | | | | | |
| Low | 34 (13.4) | 1 (5.0) | 35 (13.5) | 0 | 34 (13.9) | 1 (3.6) | 32 (15.5) | 3 (4.5) | 34 (13.1) | 1 (7.1) |
| Moderate | 105 (41.5) | 11 (55.0) | 110 (42.5) | 6 (42.9) | 101 (41.2) | 15 (53.6) | 89 (43.0) | 27 (40.9) | 108 (41.7) | 8 (57.1) |
| High | 114 (45.1) | 8 (40.0) | 114 (44.0) | 8 (57.1) | 110 (44.9) | 12 (42.9) | 86 (41.5) | 36 (54.5) | 117 (45.2) | 5 (35.7) |
| $\chi^2$; $p$ | 1.930; 0.381 | | 2.405; 0.301 | | 3.007; 0.222 | | 6.593; 0.037* | | 1.383; 0.501 | |
| **MPAM Interest Category** | | | | | | | | | | |
| Low | 55 (21.7) | 8 (40.0) | 59 (22.8) | 4 (28.6) | 51 (20.8) | 12 (42.9) | 44 (21.3) | 19 (28.8) | 56 (21.6) | 7 (50.0) |
| Medium | 124 (49.0) | 12 (60.0) | 129 (49.8) | 7 (50.0) | 122 (49.8) | 14 (50.0) | 109 (52.7) | 27 (40.9) | 131 (50.6) | 5 (35.7) |
| High | 74 (29.2) | 0 | 71 (27.4) | 3 (21.4) | 72 (29.4) | 2 (7.1) | 54 (26.1) | 20 (30.3) | 72 (27.8) | 2 (14.3) |
| $\chi^2$; $p$ | 8.977; 0.011* | | 0.369; 0.832 | | 9.877; 0.007** | | 2.945; 0.229 | | 6.119; 0.047* | |
| **MPAM Competence Category** | | | | | | | | | | |
| Low | 52 (20.6) | 7 (35.0) | 55 (21.2) | 4 (28.6) | 48 (19.6) | 11 (39.3) | 43 (20.8) | 16 (24.2) | 53 (20.5) | 6 (42.9) |
| Medium | 138 (54.5) | 12 (60.0) | 143 (55.2) | 7 (50.0) | 136 (55.5) | 14 (50.0) | 118 (57.0) | 32 (48.5) | 144 (55.6) | 6 (42.9) |
| High | 63 (24.9) | 1 (5.0) | 61 (23.6) | 3 (21.4) | 61 (24.9) | 3 (10.7) | 46 (22.2) | 18 (27.2) | 62 (23.9) | 2 (14.3) |
| $\chi^2$; $p$ | 5.022; 0.081 | | 0.422; 0.81 | | 6.805; 0.033* | | 1.484; 0.476 | | 4.002; 0.135 | |
| **MPAM Appearance Category** | | | | | | | | | | |
| Low | 63 (24.9) | 7 (35.0) | 63 (24.3) | 7 (50.0) | 59 (24.1) | 11 (39.3) | 55 (26.6) | 15 (22.7) | 66 (25.5) | 4 (28.6) |
| Medium | 121 (47.8) | 9 (45.0) | 126 (48.6) | 4 (28.6) | 119 (48.6) | 11 (39.3) | 98 (47.3) | 32 (48.5) | 122 (47.1) | 8 (57.1) |
| High | 69 (27.3) | 4 (20.0) | 70 (27.0) | 3 (21.4) | 67 (27.3) | 6 (21.4) | 54 (26.1) | 19 (28.8) | 71 (27.4) | 2 (14.3) |
| $\chi^2$; $p$ | 1.135; 0.567 | | 4.695; 0.096 | | 3.050; 0.218 | | 0.438; 0.803 | | 1.186; 0.553 | |
| **MPAM Fitness Category** | | | | | | | | | | |
| Low | 59 (23.3) | 9 (45.0) | 60 (23.2) | 8 (57.1) | 56 (22.9) | 12 (42.9) | 53 (25.6) | 15 (22.7) | 64 (24.7) | 4 (28.6) |
| Medium | 128 (50.6) | 9 (45.0) | 133 (51.4) | 4 (28.5) | 125 (51.0) | 12 (42.9) | 102 (49.3) | 35 (53.0) | 129 (49.8) | 8 (57.1) |
| High | 66 (26.1) | 2 (10.0) | 66 (25.5) | 2 (14.3) | 64 (26.1) | 4 (14.3) | 52 (25.1) | 16 (24.2) | 66 (25.5) | 2 (14.3) |
| $\chi^2$; $p$ | 5.539; 0.063 | | 8.198; 0.017* | | 5.782; 0.056 | | 0.322; 0.851 | | 0.890; 0.641 | |
| **MPAM Social Category** | | | | | | | | | | |
| Low | 58 (22.9) | 4 (20.0) | 57 (22.0) | 5 (35.7) | 56 (22.9) | 6 (21.4) | 49 (23.7) | 13 (19.7) | 59 (22.8) | 3 (21.4) |
| Medium | 137 (54.2) | 12 (60.0) | 142 (54.8) | 7 (50.0) | 131 (53.5) | 18 (64.3) | 111 (53.6) | 38 (57.6) | 139 (53.7) | 10 (71.4) |
| High | 58 (22.9) | 4 (20.0) | 60 (23.2) | 2 (14.3) | 58 (23.7) | 4 (14.3) | 47 (22.7) | 15 (22.7) | 61 (23.6) | 1 (7.1) |
| $\chi^2$; $p$ | 0.256; 0.88 | | 1.617; 0.446 | | 1.536; 0.464 | | 0.491; 0.782 | | 2.353; 0.308 | |
| **Physical Fitness Category** | | | | | | | | | | |
| Good | 28 (11.1) | 1 (5.0) | 28 (10.8) | 1 (7.1) | 27 (11.0) | 2 (7.1) | 23 (11.1) | 6 (9.1) | 28 (10.8) | 1 (7.1) |
| Average | 26 (10.3) | 0 | 24 (9.3) | 2 (14.3) | 25 (10.2) | 1 (3.6) | 18 (8.7) | 8 (12.1) | 26 (10.0) | 0 |
| Poor | 199 (78.7) | 19 (95.0) | 207 (79.9) | 11 (78.6) | 193 (78.8) | 25 (89.3) | 166 (80.2) | 52 (78.8) | 205 (79.2) | 13 (92.9) |
| $\chi^2$; $p$ | 3.318; 0.19 | | 0.523; 0.77 | | 1.864; 0.394 | | 0.821; 0.663 | | 1.886; 0.389 | |

Parentheses indicate percentages out of the same anthropometric class.

BMI: Body Mass Index; WC: Waist Circumference; WHR: Waist-Hip Ratio; WHtR: Waist-Height Ratio; MPAM: Motives for Physical Activities Measure.

*$p$-value is significant at the 0.05 level (2-tailed); **$p$-value is significant at the 0.01 level (2-tailed).

motive. Higher levels of PA were linked with higher intrinsic motivations [32], which reflects on our result where those who showed higher PA levels (men) also showed higher intrinsic motivation. Of note, although not significantly different from each other, both genders scored the highest for Fitness scores among all five intrinsic motivation categories, which is supported by past Malaysian studies [31,33].

**Table 4. Association of _OXTR_ rs53576 and rs2254298 genotypes with demographics, PA level, intrinsic exercise motivation, physical fitness and anthropometric and body composition classes.**

| | rs53576 Genotypes | | | rs2254298 Genotypes | | |
|---|---|---|---|---|---|---|
| | AA (_n_=119) | AG (_n_=114) | GG (_n_=40) | GG (_n_=106) | GA (_n_=90) | AA (_n_=77) |
| **Gender** | | | | | | |
| Male | 50 (42.4) | 55 (46.6) | 13 (11.0) | 44 (37.3) | 40 (33.9) | 34 (28.8) |
| Female | 69 (44.5) | 59 (38.1) | 27 (17.4) | 62 (40.0) | 50 (32.3) | 43 (27.7) |
| $\chi^2$; _p_ | 3.117; 0.21 | | | 0.209; 0.901 | | |
| **Ethnicity** | | | | | | |
| Malay/Bumiputra | 2 (12.5) | 11 (68.8) | 3 (18.8) | 6 (37.5) | 4 (25.0) | 6 (37.5) |
| Chinese | 114 (49.8) | 87 (38.0) | 28 (12.2) | 84 (36.7) | 77 (33.6) | 68 (29.7) |
| Indian | 3 (10.7) | 16 (57.1) | 9 (32.1) | 16 (57.1) | 9 (32.1) | 3 (10.7) |
| $\chi^2$; _p_ | 24.610; <0.001** | | | 6.749; 0.15 | | |
| **Total PA Level** | | | | | | |
| Low | 13 (37.1) | 15 (42.9) | 7 (20.0) | 16 (45.7) | 7 (20.0) | 12 (34.3) |
| Moderate | 61 (52.6) | 42 (36.2) | 13 (11.2) | 49 (42.2) | 34 (29.3) | 33 (28.4) |
| High | 45 (36.9) | 57 (46.7) | 20 (16.4) | 41 (33.6) | 49 (40.2) | 32 (26.2) |
| $\chi^2$; _p_ | 7.207; 0.125 | | | 6.435; 0.169 | | |
| **MPAM Interest Category** | | | | | | |
| Low | 28 (44.4) | 25 (39.7) | 10 (15.9) | 22 (34.9) | 20 (31.7) | 21 (33.3) |
| Medium | 59 (43.4) | 58 (42.6) | 19 (14.0) | 58 (42.6) | 43 (31.6) | 35 (25.7) |
| High | 32 (43.2) | 31 (41.9) | 11 (14.9) | 26 (35.1) | 27 (36.5) | 21 (28.4) |
| $\chi^2$; _p_ | 0.214; 0.995 | | | 2.282; 0.684 | | |
| **MPAM Competence Category** | | | | | | |
| Low | 24 (40.7) | 24 (40.7) | 11 (18.6) | 23 (39.0) | 18 (30.5) | 18 (30.5) |
| Medium | 72 (48.0) | 61 (40.7) | 17 (11.3) | 58 (38.7) | 51 (34.0) | 41 (27.3) |
| High | 23 (35.9) | 29 (45.3) | 12 (18.8) | 25 (39.1) | 21 (32.8) | 18 (28.1) |
| $\chi^2$; _p_ | 4.400; 0.355 | | | 0.311; 0.989 | | |
| **MPAM Appearance Category** | | | | | | |
| Low | 26 (37.1) | 35 (50.0) | 9 (12.9) | 29 (41.4) | 20 (28.6) | 21 (30.0) |
| Medium | 67 (51.5) | 49 (37.7) | 14 (10.8) | 48 (36.9) | 48 (36.9) | 34 (26.2) |
| High | 26 (35.6) | 30 (41.1) | 17 (23.3) | 29 (39.7) | 22 (30.1) | 22 (30.1) |
| $\chi^2$; _p_ | 10.484; 0.033* | | | 1.834; 0.766 | | |
| **MPAM Fitness Category** | | | | | | |
| Low | 29 (42.6) | 29 (42.6) | 10 (14.7) | 28 (41.2) | 21 (30.9) | 19 (27.9) |
| Medium | 63 (46.0) | 55 (40.1) | 19 (13.9) | 50 (36.5) | 46 (33.6) | 41 (29.9) |
| High | 27 (39.7) | 30 (44.1) | 11 (16.2) | 28 (41.2) | 23 (33.8) | 17 (25.0) |
| $\chi^2$; _p_ | 0.784; 0.941 | | | 0.899; 0.925 | | |
| **MPAM Social Category** | | | | | | |
| Low | 29 (46.8) | 23 (37.1) | 10 (16.1) | 26 (41.9) | 23 (37.1) | 13 (21.0) |
| Medium | 62 (41.6) | 66 (44.3) | 21 (14.1) | 56 (37.6) | 44 (29.5) | 49 (32.9) |
| High | 28 (45.2) | 25 (40.3) | 9 (14.5) | 24 (38.7) | 23 (37.1) | 15 (24.2) |
| $\chi^2$; _p_ | 1.021; 0.907 | | | 4.052; 0.399 | | |
| **Physical Fitness Category** | | | | | | |
| Good | 12 (41.4) | 11 (37.9) | 6 (20.7) | 12 (41.4) | 11 (37.9) | 6 (20.7) |
| Average | 13 (50.0) | 11 (42.3) | 2 (7.7) | 9 (34.6) | 8 (30.8) | 9 (34.6) |
| Poor | 94 (43.1) | 92 (42.2) | 32 (14.7) | 85 (39.0) | 71 (32.6) | 62 (28.4) |
| $\chi^2$; _p_ | 1.984; 0.739 | | | 1.398; 0.845 | | |

_(Continued)_

| | rs53576 Genotypes | | | rs2254298 Genotypes | | |
|---|---|---|---|---|---|---|
| | AA (*n*=119) | AG (*n*=114) | GG (*n*=40) | GG (*n*=106) | GA (*n*=90) | AA (*n*=77) |
| **Blood Pressure Category** | | | | | | |
| Normal | 94 (42.3) | 94 (42.3) | 34 (15.3) | 84 (37.8) | 72 (32.4) | 66 (29.7) |
| Prehypertension | 22 (47.8) | 18 (39.1) | 6 (13.0) | 20 (43.5) | 16 (34.8) | 10 (21.7) |
| Stage 1 hypertension | 2 (50.0) | 2 (50.0) | 0 | 1 (25.0) | 2 (50.0) | 1 (25.0) |
| Stage 2 hypertension | 1 (100) | 0 | 0 | 1 (100) | 0 | 0 |
| $\chi^2$; *p* | 2.494; 0.869 | | | 3.381; 0.76 | | |
| **WC Class** | | | | | | |
| Normal | 110 (43.5) | 104 (41.1) | 39 (15.4) | 100 (39.5) | 80 (31.6) | 73 (28.9) |
| High | 9 (45.0) | 10 (50.0) | 1 (5.0) | 6 (30.0) | 10 (50.0) | 4 (20.0) |
| $\chi^2$; *p* | 1.733; 0.42 | | | 2.847; 0.241 | | |
| **WHR Class** | | | | | | |
| Normal | 115 (44.4) | 106 (40.9) | 38 (14.7) | 102 (39.4) | 83 (32.0) | 74 (28.6) |
| High | 4 (28.6) | 8 (57.1) | 2 (14.3) | 4 (28.6) | 7 (50.0) | 3 (21.4) |
| $\chi^2$; *p* | 1.601; 0.449 | | | 1.939; 0.379 | | |
| **WHtR Class** | | | | | | |
| Normal | 108 (44.1) | 99 (40.4) | 38 (15.5) | 95 (38.8) | 79 (32.2) | 71 (29.0) |
| High | 11 (39.2) | 15 (53.6) | 2 (7.1) | 11 (39.3) | 11 (39.3) | 6 (21.4) |
| $\chi^2$; *p* | 2.376; 0.305 | | | 0.888; 0.642 | | |
| **TBF Class** | | | | | | |
| Normal | 87 (47.5) | 71 (38.8) | 25 (13.7) | 68 (37.2) | 60 (32.8) | 55 (30.1) |
| High | 32 (35.6) | 43 (47.8) | 15 (16.7) | 38 (42.2) | 30 (33.3) | 22 (24.4) |
| $\chi^2$; *p* | 3.525; 0.172 | | | 1.077; 0.584 | | |
| **VFL Class** | | | | | | |
| Normal | 112 (43.8) | 105 (41.0) | 39 (15.2) | 99 (38.7) | 85 (33.2) | 72 (28.1) |
| High | 7 (41.2) | 9 (52.9) | 1 (5.9) | 7 (41.2) | 5 (29.4) | 5 (29.4) |
| $\chi^2$; *p* | 1.519; 0.468 | | | 0.105; 0.949 | | |
| **BMI Overweight Class** | | | | | | |
| Normal | 94 (45.4) | 81 (39.1) | 32 (15.5) | 81 (39.1) | 64 (30.9) | 62 (30.0) |
| Overweight | 25 (37.9) | 33 (50.0) | 8 (12.1) | 25 (37.9) | 26 (39.4) | 15 (22.7) |
| $\chi^2$; *p* | 2.448; 0.294 | | | 2.037; 0.361 | | |
| **BMI Obese Class** | | | | | | |
| Normal | 112 (43.2) | 108 (41.7) | 39 (15.1) | 101 (39.0) | 86 (33.2) | 72 (27.8) |
| Obese | 7 (50.0) | 6 (42.9) | 1 (7.1) | 5 (35.7) | 4 (28.6) | 5 (35.7) |
| $\chi^2$; *p* | 0.711; 0.701 | | | 0.418; 0.811 | | |
| **SM Class** | | | | | | |
| Normal | 72 (41.9) | 74 (43.0) | 26 (15.1) | 72 (41.9) | 57 (33.1) | 43 (25.0) |
| High | 47 (46.5) | 40 (39.6) | 14 (13.9) | 34 (33.7) | 33 (32.7) | 34 (33.7) |
| $\chi^2$; *p* | 0.565; 0.754 | | | 2.799; 0.247 | | |

Parentheses indicate percentages out of the same demographic, PA level, intrinsic exercise motivation, physical fitness and anthropometric/body composition class.

BMI: Body Mass Index; TBF: Total Body Fat; VFL: Visceral Fat Level; SM: Skeletal Muscle Percentage; WC: Waist Circumference; WHR: Waist-Hip Ratio; WHtR: Waist-Height Ratio; MPAM: Motives for Physical Activities Measure.

*\*p*-value is significant at the 0.05 level (2-tailed); *\*\*p*-value is significant at the 0.01 level (2-tailed).

Out of the five intrinsic motivation categories, only Interest and Competence were significantly correlated with PA levels and physical fitness. Higher Interest and Competence were correlated with higher engagement hours in vigorous, moderate, and total PA, but were correlated with lower engagement hours in mundane physical activity (walking) and sedentary activity (sitting). This finding is consistent with previous studies [27,28,33–35], which similarly found that both of these motives were important predictive factors for exercise participation. Higher general exercise intrinsic motivation, but not identified, introjected or external motivation, was associated with higher physical activity but lower sitting time [36]. The reason is that higher enjoyment predicts higher participation, desire, commitment, and adherence in sports [37], whereas competence gives an individual a competitive advantage during tournaments/matches [27]. The finding that higher Interest and Competence led to lower physical fitness might be spurious, confounded by the fact that 80% of the participants were categorised as having poor PA level. In contrast, exercise intrinsic motivation of Chinese college students was found to be directly related to physical fitness, or indirectly related through the mediating effect of PA [32].

In our study, PA levels were not significantly associated with most anthropometric and body composition classifications, either among overall participants or within genders. Nevertheless, our study also observed that a proportion of respondents with high PA levels were found within the overweight group. The presence of physically active individuals within the higher BMI groups could be attributed to various factors. For example, a higher proportion of muscle mass could contribute to their BMI while maintaining a healthy PA level. Higher PA levels among overweight adults were also reflected in the 2011 Malaysian National Health and Morbidity Survey, involving 10,141 individuals [38]. Furthermore, our study demonstrated a noteworthy association between PA levels and VFL among men, consistent with a meta-analysis that found that men benefited more from exercise in terms of VFL reduction compared to women [39]. In terms of exercise intrinsic motivation, we found that those who were obese or had high central adiposity (as indicated by high WC and WHtR) showed lower Interest, Competence, or Fitness scores. This is consistent with previous studies which showed that non-obese had higher exercise intrinsic motivation scores than obese [40,41].

The rs53576 and rs2254298 gene variants are located non-coding third intron of OXTR and would not affect the expression of the OXTR protein. A search in the VarSome database for rs53576 and rs2254298 (https://varsome.com/variant/hg19/rs53576? and https://varsome.com/variant/hg19/rs2254298?, respectively) revealed a Deleterious Annotation of genetic variants using Neural Networks (DANN) scores of 0.6058 and 0.3451, indicating benign moderate and benign strong damaging effects of the SNP, respectively. Nevertheless, these SNPs are located in the intron containing OCE7, a cis-regulatory element that has a robust enhancer activity for Oxtr gene in mouse hypothalamus cells [42], suggesting potential effects on behavioural phenotypes in humans. To the best of our knowledge, the association between OXTR rs53576 and rs2254298 with PA levels, exercise intrinsic motivation and physical fitness has not been investigated yet. Although OXTR gene variants have primarily been studied in relation to social and emotional traits [9], their influence on PA is biologically plausible. Oxytocin interacts with the brain's reward and motivation systems, particularly the dopaminergic pathway, which may affect how individuals experience intrinsic rewards from PA [14]. Since intrinsic motivation is shaped by enjoyment, competence, and social connection—domains where oxytocin is active—OXTR gene variants may help explain individual differences in PA behaviour. While this genetic link remains underexplored, especially in non-Western populations, examining OXTR in relation to PA and motivation offers a novel, biologically grounded perspective to understanding PA engagement. In this study, we found that their MAFs among overall participants were 0.36 and 0.45, respectively, consistent with MAFs among all Asians in the 1000Genomes Project [43–45]. Of note, the frequency of A allele of rs53376 differs a lot between the world populations – Europeans and Africans demonstrated comparable A allele frequencies, and in Asian populations its frequencies were much higher [45]. In regards to rs2254298, in Europeans, particularly of northern origin, the frequency of this allele is minimal, while in Asian populations the A allele frequency is much higher [45]. Africans demonstrate intermediate frequencies but closer to Asian ones [45]. In this study, we found that rs53576 genotype was only significantly with Appearance intrinsic

motivation, but not with other categories. The Appearance intrinsic motivation refers to motivation for being physically active in order to become more physically attractive, to have defined muscles, to look better, and to achieve or maintain a desired weight [6]. Since the oxytocin system is involved in sexual- and social-related behaviours, the positive association between rs53576 and Appearance is therefore justifiable. Those carrying the rs53576 GG genotype or G allele had higher intrinsic exercise motivation to look more physically attractive, possibly to have better sexual and social appeal. Adiposity-wise, those carrying the rs53576 heterozygous AG and homozygous variant GG genotypes had higher obesity risk among Turkish adolescents [13]. Related to obesity, a study found that the two *OXTR* gene variants were associated with eating disorders – the rs53576 A allele negatively correlated to binging/purging behaviours, GG genotype was at increased risk of engaging in binging/purging, while rs2254298 A allele carriers were at increased odds of restrictive eating/purging [46]. However, we found no association between these two *OXTR* gene variants, possibly due to ethnic population differences and a small sample size which limits the statistical power for genetic analyses. Finally, we also found no association between rs53576 and rs2254298 with PA levels and physical fitness. Nevertheless, a previous study found that the rs53576 genotypes significantly modulated the correlation between total MET-min/wk, walking MET-min/wk and sitting time min/d with cognitive empathy score, and between vigorous MET-min/wk with affective empathy score [47].

There are a few limitations in this study. First, the results of this study are not generalizable or representative of all the young urban adults in Malaysia. This is because the participants were only recruited from one single university and were also ethnically imbalanced, where most participants were Chinese. A larger and more ethnically-balanced sample size is foreseeable in the future. Second, the measures used in this study were self-reported, indicating the possibility of recall bias or social desirability bias. This means participants might recall wrongly the actual amount of time spent doing PA or may purposely under/over-report their PA to appear more favourable. To complement self-reported measures, accelerometer or activity tracker bands could be used to objectively track physical activities. Third, as this study was cross-sectional we were not able to capture trends or changes in the participants over time. A longitudinal study could be conducted to observe changes and maintenance in PA levels, intrinsic motivation, physical fitness, and adiposity when transitioning between different stages of life. Lastly, the association between other forms of motivations (e.g., extrinsic motives, amotivation) and PA levels was not investigated in this study. Future studies with larger sample sizes and longitudinal designs may benefit from the use of Structural Equation Modelling to better model mediating or moderating pathways between genetic, psychological, and behavioural variables.

## Conclusions

The main aim of this research was to study the association between the motives and levels of PA among a sample of young urban Malaysian adults. The significance of this study includes helping us to gain a deeper understanding of the intrinsic motives that encourage young adults to participate in PA, i.e., interest/enjoyment and competence, and how these motives can affect PA levels. This study also highlighted the gender differences in PA participation motives and levels, where women reported lesser motivation and were less active than men. Not only that, this study also highlighted the importance of PA in maintaining healthy physical fitness, as well as how the lack of it can be detrimental. These findings can ultimately help us create effective interventions to increase PA participation among the targeted groups and hence, build a healthier community of tertiary students in Malaysia, which is in line with the Sustainable Development Goal 3 (Good Health and Well-Being).

## Supporting information

**S1 Table. Association of OXTR rs53576 and rs2254298 alleles with demographics, PA level, intrinsic exercise motivation, physical fitness, and anthropometric and body composition classes.**
(DOCX)

## Acknowledgments

The authors would like to thank all participants for participating in this study. We would also like to thank Alex Ern Che Lee for his assistance in performing the measurements and fitness test.

## Author contributions

**Conceptualization:** Yee-How Say.

**Data curation:** Kristin-Ann Zhe Mun Leong, Hui Wen Ng.

**Formal analysis:** Yee-How Say, Kristin-Ann Zhe Mun Leong, Hui Wen Ng, Zi Di Ng.

**Investigation:** Kristin-Ann Zhe Mun Leong, Hui Wen Ng, Zi Di Ng.

**Methodology:** Yee-How Say, Jack Bee Chook.

**Supervision:** Yee-How Say, Geetha Letchumanan, Jack Bee Chook.

**Validation:** Yee-How Say, Kristin-Ann Zhe Mun Leong, Hui Wen Ng, Geetha Letchumanan, Jack Bee Chook.

**Visualization:** Yee-How Say, Kristin-Ann Zhe Mun Leong, Hui Wen Ng, Geetha Letchumanan, Jack Bee Chook.

**Writing – original draft:** Yee-How Say, Kristin-Ann Zhe Mun Leong, Hui Wen Ng, Geetha Letchumanan.

**Writing – review & editing:** Yee-How Say, Jack Bee Chook.

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
