## [Decision Letter · Decision Letter 0]

22 Jul 2025

PONE-D-25-11325Physical activity levels, exercise intrinsic motivation, physical fitness, and their association with adiposity and Oxytocin Receptor (OXTR) rs53576 and rs2254298 gene variants among Malaysian urban young adultsPLOS ONE

Dear Dr. Say,

Thank you for submitting your manuscript to PLOS ONE. After careful consideration, we feel that it has merit but does not fully meet PLOS ONE’s publication criteria as it currently stands. Therefore, we invite you to submit a revised version of the manuscript that addresses the points raised during the review process.

There appear to be both conceptual and writing issues in your manuscript. Please carefully review the comments provided and revise the manuscript thoroughly to address the concerns raised.

We look forward to receiving your revised manuscript.

Kind regards,

Zohreh Sajadi Hezaveh

Academic Editor

PLOS ONE

Journal Requirements:

3. In the online submission form, you indicated that data will be made available on request.

The authors would like to thank all participants for participating in this study. We would also like to thank Alex Ern Che Lee for his assistance is performing the measurements and fitness test. This research is supported by the Malaysian Ministry of Higher Education Fundamental Grant Research Scheme FRGS/1/2022/STG01/SYUC/02/1. The funder had no role in the design of the study; in the collection, analyses, or interpretation of data; in the writing of the manuscript; or in the decision to publish the results.

This research is supported by the Malaysian Ministry of Higher Education Fundamental Grant Research Scheme FRGS/1/2022/STG01/SYUC/02/1. The funder had no role in the design of the study; in the collection, analyses, or interpretation of data; in the writing of the manuscript; or in the decision to publish the results.

Reviewers' comments:

Reviewer's Responses to Questions

**Comments to the Author**

1. Is the manuscript technically sound, and do the data support the conclusions?

Reviewer #1: Partly

Reviewer #2: Partly

2. Has the statistical analysis been performed appropriately and rigorously? 

Reviewer #1: Yes

Reviewer #2: Yes

3. Have the authors made all data underlying the findings in their manuscript fully available?

Reviewer #1: No

Reviewer #2: Yes

4. Is the manuscript presented in an intelligible fashion and written in standard English?

Reviewer #1: Yes

Reviewer #2: Yes

5. Review Comments to the Author

Reviewer #1: The manuscript presents an intriguing approach by correlating physical fitness parameters and engagement in sports activities with genetic polymorphisms, notably contributing to a growing field of research that seeks to elucidate the biological underpinnings of behavioral tendencies related to exercise. This association may, in the long term, enhance our understanding of individual predispositions toward physical activity and support the development of population-level strategies aimed at promoting exercise adherence — a particularly timely contribution given the alarming global increase in obesity rates and the recognized importance of physical activity in its prevention and management.

Nevertheless, several limitations within the study merit consideration:

The authors are encouraged to perform interaction analyses between the investigated variables and the identified SNPs, in order to more robustly assess the strength and nature of these associations.

Although the manuscript states that the SNPs in question do not alter oxytocin protein levels, it does not provide mechanistic insight into how these polymorphisms might impact downstream signaling pathways. It is recommended that the authors incorporate in silico or in vitro analyses to investigate potential structural alterations in the oxytocin receptor and their functional implications for signal transduction.

Furthermore, the employment of genetically engineered experimental models bearing the same or analogous mutations in the OXTR gene would offer a valuable opportunity to explore the molecular mechanisms by which such polymorphisms influence motivation and behavioral responses to physical exercise.

Lastly, it is suggested that the title of the manuscript be revised to reduce its regional specificity and to more effectively underscore the broader relevance of the observed associations between genetic polymorphisms and the phenotypic traits under investigation.

Reviewer #2: Major Scientific and Conceptual Issues:

1. Lack of Theoretical Integration:

The manuscript reads like multiple unrelated studies combined into one: a PA/motivation study, a fitness/adiposity study, and a genetics study. The relationships between these variables are not clearly explained, and no cohesive theoretical model is presented to justify their joint analysis. The importance and applicability of studying these variables together remain unclear.

2. Overly Long and Unfocused Introduction:

The introduction contains excessive background information on common knowledge (e.g., definitions of motivation types). It lacks coherence, jumping from topic to topic without clearly connecting them. Although gaps are mentioned (e.g., lack of studies on motivation and PA, inconsistent gender findings, absence of Malaysian data), their significance is not convincingly justified.

3. Redundancy in Methodology:

The methods section includes obvious and unnecessary statements, such as:“Participants who did not fulfil the inclusion criteria were excluded from the study.” Such redundancy detracts from the clarity and professionalism of the methods.

4. Insufficient Power for Genetic Analyses:

No power calculation is provided for the genetic component. Given the modest sample size (N=273), the study may be underpowered to detect meaningful genetic associations.

5. Weak Justification for Genetic Focus:

The study claims to be the first to assess OXTR variants with PA and motivation, but the biological mechanism linking these variables is unclear. Prior associations of OXTR are mostly in the context of social, emotional, or psychiatric traits, not physical activity behavior.

6. Use of Self-Reported PA Measures:

PA is assessed using the self-reported IPAQ-SF, which is prone to recall and social desirability bias. Drawing associations between self-reported PA and biological/genetic variables is questionable without objective PA data (e.g., accelerometry).

7. Missing Methodological References:

No reference is provided for the genotyping protocol used. No citation is given for the interpretation of correlation coefficient thresholds either (e.g., what defines weak, moderate, strong).

8. Causal Language in Discussion of Cross-Sectional Data:

Although the authors note the study is cross-sectional, the discussion includes causal statements (e.g., “intrinsic motivation leads to more PA, which reduces adiposity”), which are inappropriate for the study design.

9. Ambiguity in Socioeconomic Classification:

The term “M40” is used without explanation. What does it stand for?

6. PLOS authors have the option to publish the peer review history of their article (what does this mean? ). If published, this will include your full peer review and any attached files.

**Do you want your identity to be public for this peer review?** For information about this choice, including consent withdrawal, please see our Privacy Policy .

Reviewer #1: No

Reviewer #2: No

---

## [Author Response · Author response to Decision Letter 1]

24 Jul 2025

Physical activity levels, exercise intrinsic motivation, physical fitness, and their association with adiposity and Oxytocin Receptor (OXTR) rs53576 and rs2254298 gene variants among Malaysian urban young adults

PONE-D-25-11325

PLOS One

We sincerely thank you and the reviewers for carefully reading our manuscript and providing constructive comments, and we have addressed them point-by-point and amended the manuscript, as follows:

Comments Response to Reviewers

Editor

• Please ensure that your manuscript meets PLOS ONE's style requirements, including those for file naming. • Manuscript amended to fulfil PLOS ONE’s style requirements.

• Your ethics statement should only appear in the Methods section of your manuscript. If your ethics statement is written in any section besides the Methods, please delete it from any other section. • Ethical statement deleted from declarations section.

• In the online submission form, you indicated that data will be made available on request.

All PLOS journals now require all data underlying the findings described in their manuscript to be freely available to other researchers, either a. In a public repository, b. Within the manuscript itself, or c. Uploaded as supplementary information. • Data availability amended to “All relevant data are within the manuscript.”

• We note that you have included the phrase “data not shown” in your manuscript. Unfortunately, this does not meet our data sharing requirements. PLOS does not permit references to inaccessible data. We require that authors provide all relevant data within the paper, Supporting Information files, or in an acceptable, public repository. Please add a citation to support this phrase or upload the data that corresponds with these findings to a stable repository (such as Figshare or Dryad) and provide and URLs, DOIs, or accession numbers that may be used to access these data. Or, if the data are not a core part of the research being presented in your study, we ask that you remove the phrase that refers to these data. • Phrases “data not shown” removed. Additional data has been added as Supporting Information S1 Table.

Please remove any funding-related text from the manuscript and let us know how you would like to update your Funding Statement. • Funding statement removed from Acknowledgements. Declaration of Funding Statement in the online submission form has been amended to “This research is supported by the Malaysian Ministry of Higher Education Fundamental Grant Research Scheme FRGS/1/2022/STG01/SYUC/02/1. The funder had no role in the design of the study; in the collection, analyses, or interpretation of data; in the writing of the manuscript; or in the decision to publish the results.”

Reviewer 1

• The authors are encouraged to perform interaction analyses between the investigated variables and the identified SNPs, in order to more robustly assess the strength and nature of these associations. Although Structural Equation Modelling (SEM) is a powerful technique for examining complex, multi-pathway relationships among latent and observed variables, it was not employed in this study due to both methodological and practical considerations. First, the primary aim of this study was to examine direct associations and group differences among measured variables such as PA levels, intrinsic motivation subscales, physical fitness, adiposity, and OXTR genotypes. These objectives were more appropriately addressed using conventional statistical methods such as correlation, regression, and group comparisons.

Second, SEM requires large sample sizes to yield stable and valid estimates, particularly when models include multiple latent constructs and pathways. Given the modest sample size of this study and the inclusion of genetic data with categorical variants, SEM would have lacked the statistical power necessary to produce reliable model fit and parameter estimates. Moreover, this study focused on exploratory rather than confirmatory modelling, making simpler, theory-informed analyses more suitable at this stage.

Future studies with larger sample sizes and longitudinal designs may benefit from the use of SEM to better model mediating or moderating pathways between genetic, psychological, and behavioral variables.

The last point above has also been added to the limitations of study/future studies paragraph in Discussion.

• Although the manuscript states that the SNPs in question do not alter oxytocin protein levels, it does not provide mechanistic insight into how these polymorphisms might impact downstream signaling pathways. It is recommended that the authors incorporate in silico or in vitro analyses to investigate potential structural alterations in the oxytocin receptor and their functional implications for signal transduction. In silico prediction of the effect of SNPs like SIFT, PolyPhen2, SNAp, PhD-SNP all are irrelevant, as the SNPs are not missense (protein-coding) SNPs.

SNPinfo search (https://snpinfo.niehs.nih.gov/cgi-bin/snpinfo/snpfunc.cgi) revealed no results for the SNPs.

A search in the VarSome database for rs53576 and rs2254298 (https://varsome.com/variant/hg19/rs53576? and https://varsome.com/variant/hg19/rs2254298?, respectively) revealed Deleterious Annotation of genetic variants using Neural Networks (DANN) scores of 0.6058 and 0.3451, indicating benign moderate and benign strong damaging effects of the SNP, respectively.

These SNPs are located in the intron containing OCE7, a cis-regulatory element (cCRE) that has a robust enhancer activity for Oxtr gene in mouse hypothalamus cells (Laboy et al., 2025), suggesting potential effects on behavioural phenotypes in humans.

• Furthermore, the employment of genetically engineered experimental models bearing the same or analogous mutations in the OXTR gene would offer a valuable opportunity to explore the molecular mechanisms by which such polymorphisms influence motivation and behavioral responses to physical exercise. These SNPs are located in the intron containing OCE7, a cis-regulatory element (cCRE) that has a robust enhancer activity for Oxtr gene in mouse hypothalamus cells (Laboy et al., 2025), suggesting potential effects on behavioural phenotypes in humans.

• Lastly, it is suggested that the title of the manuscript be revised to reduce its regional specificity and to more effectively underscore the broader relevance of the observed associations between genetic polymorphisms and the phenotypic traits under investigation. Title amended to “Physical activity levels, exercise intrinsic motivation, physical fitness, and their association with adiposity and Oxytocin Receptor (OXTR) rs53576 and rs2254298 gene variants”

Reviewer 2

• Lack of Theoretical Integration:

The manuscript reads like multiple unrelated studies combined into one: a PA/motivation study, a fitness/adiposity study, and a genetics study. The relationships between these variables are not clearly explained, and no cohesive theoretical model is presented to justify their joint analysis. The importance and applicability of studying these variables together remain unclear. Introduction has been rewritten to show interrelatedness between the multiple components of PA/motivation study, a fitness/adiposity study, and a genetics study.

• Overly Long and Unfocused Introduction:

The introduction contains excessive background information on common knowledge (e.g., definitions of motivation types). It lacks coherence, jumping from topic to topic without clearly connecting them. Although gaps are mentioned (e.g., lack of studies on motivation and PA, inconsistent gender findings, absence of Malaysian data), their significance is not convincingly justified. Introduction has been rewritten to be shortened, more focused, and with the significance of study more justified.

• Redundancy in Methodology:

The methods section includes obvious and unnecessary statements, such as: “Participants who did not fulfil the inclusion criteria were excluded from the study.” Such redundancy detracts from the clarity and professionalism of the methods. The sentence “Participants who did not fulfil the inclusion criteria were excluded from the study” is deleted.

• Insufficient Power for Genetic Analyses:

No power calculation is provided for the genetic component. Given the modest sample size (N=273), the study may be underpowered to detect meaningful genetic associations. Using the Genetic Association Study Power Calculator (https://csg.sph.umich.edu/abecasis/cats/gas_power_calculator/), the statistical power for rs53576 and rs2254298 SNPs were 80% and 50%, respectively, assuming that overweight is the “disease” or outcome; the case/control sample size of 207 normal and 66 overweight; the significance level of 0.05; the disease model is additive; the prevalence of overweight among Malaysians is 31.3% [24]; the disease allele frequency is 0.36 and 0.45 for rs53576 and rs2254298, respectively; and the heterozygous genotype relative risk is 1.532 and 1.316, respectively Due to the weak statistical power particularly for rs2254298, this limitation has been added to the Discussion.

• Weak Justification for Genetic Focus:

The study claims to be the first to assess OXTR variants with PA and motivation, but the biological mechanism linking these variables is unclear. Prior associations of OXTR are mostly in the context of social, emotional, or psychiatric traits, not physical activity behavior. A stronger justification has been provided in Discussion as follows:

Although OXTR gene variants have primarily been studied in relation to social and emotional traits [9], their influence on PA is biologically plausible. Oxytocin interacts with the brain’s reward and motivation systems, particularly the dopaminergic pathway, which may affect how individuals experience intrinsic rewards from PA [14]. Since intrinsic motivation is shaped by enjoyment, competence, and social connection—domains where oxytocin is active—OXTR gene variants may help explain individual differences in PA behaviour. While this genetic link remains underexplored, especially in non-Western populations, examining OXTR in relation to PA and motivation offers a novel, biologically grounded perspective to understanding PA engagement.

• Use of Self-Reported PA Measures:

PA is assessed using the self-reported IPAQ-SF, which is prone to recall and social desirability bias. Drawing associations between self-reported PA and biological/genetic variables is questionable without objective PA data (e.g., accelerometry). This has already been mentioned in the list of limitations of the study as follows:

To complement self-reported measures, accelerometer or activity tracker bands could be used to objectively track physical activities.

• Missing Methodological References:

No reference is provided for the genotyping protocol used. No citation is given for the interpretation of correlation coefficient thresholds either (e.g., what defines weak, moderate, strong). Genotyping was performed according to the manufacturer’s protocols. This has been added to the Methods:

Genomic DNA was extracted from mouthwash samples using the GF-1 Tissue DNA Extraction Kit (Vivantis, Malaysia), before proceeding for genotyping of OXTR rs53576 and rs2254298 using the FRET (fluorescent resonance energy transfer) chemistry allele-specific real-time PCR-based KASP™ genotyping assay (LGC Biosearch Technologies, UK) according to the manufacturer’s protocols.

A reference for the interpretation of the correlation values has been provided: Evans J. Straightforward statistics for the behavioral sciences. Brooks/Cole Publishing; Pacific Grove, California; 1996.

• Causal Language in Discussion of Cross-Sectional Data:

Although the authors note the study is cross-sectional, the discussion includes causal statements (e.g., “intrinsic motivation leads to more PA, which reduces adiposity”), which are inappropriate for the study design. Causal language removed. Sentence changed to:

Higher Interest and Competence were correlated with higher engagement hours in vigorous, moderate, and total PA, but were correlated with lower engagement hours in mundane physical activity (walking) and sedentary activity (sitting).

• Ambiguity in Socioeconomic Classification:

The term “M40” is used without explanation. What does it stand for? The definitions of B40, M40, and T20 are already given in the footnote of Table 1.

An additional definition of M40 has also been added to paragraph 1 of Results:

…M40 monthly household income category (considered as middle-income group with the monthly income of MYR 4851-10,960)…

With the above amendments, it is our great hope that our manuscript will be accepted for publication. Thank you.

A tabulated form of the response is in the appended files.

---

## [Decision Letter · Decision Letter 1]

2 Sep 2025

Physical activity levels, exercise intrinsic motivation, physical fitness, and their association with adiposity and Oxytocin Receptor (OXTR) rs53576 and rs2254298 gene variants

PONE-D-25-11325R1

Dear Dr. Say,

We’re pleased to inform you that your manuscript has been judged scientifically suitable for publication and will be formally accepted for publication once it meets all outstanding technical requirements.

Kind regards,

Zohreh Sajadi Hezaveh

Academic Editor

PLOS ONE

Additional Editor Comments (optional):

Reviewer #1:

Reviewers' comments:

Reviewer's Responses to Questions

**Comments to the Author**

1. If the authors have adequately addressed your comments raised in a previous round of review and you feel that this manuscript is now acceptable for publication, you may indicate that here to bypass the “Comments to the Author” section, enter your conflict of interest statement in the “Confidential to Editor” section, and submit your "Accept" recommendation.

Reviewer #1: All comments have been addressed

2. Is the manuscript technically sound, and do the data support the conclusions?

Reviewer #1: Yes

3. Has the statistical analysis been performed appropriately and rigorously? 

Reviewer #1: Yes

4. Have the authors made all data underlying the findings in their manuscript fully available?

Reviewer #1: Yes

5. Is the manuscript presented in an intelligible fashion and written in standard English?

Reviewer #1: Yes

6. Review Comments to the Author

Reviewer #1: The authors have adequately addressed all the comments raised in the previous round of review. The methodological choices were justified, the limitations were clearly acknowledged, and additional clarifications and supporting information have been provided where necessary.

The manuscript is technically sound, the data support the conclusions, and the presentation is clear and intelligible. The revisions to the title and the data availability statement have improved the overall quality and compliance with journal requirements.

I have no further concerns. I recommend acceptance of the manuscript in its current form.

7. PLOS authors have the option to publish the peer review history of their article (what does this mean? ). If published, this will include your full peer review and any attached files.

**Do you want your identity to be public for this peer review?** For information about this choice, including consent withdrawal, please see our Privacy Policy .

Reviewer #1: **Yes: ** Guilherme Augusto da Silva Nogueira

---

## [Editor Report · Acceptance letter]

PONE-D-25-11325R1

PLOS ONE

Dear Dr. Say,

I'm pleased to inform you that your manuscript has been deemed suitable for publication in PLOS ONE. Congratulations! Your manuscript is now being handed over to our production team.

Kind regards,

on behalf of

Dr. Zohreh Sajadi Hezaveh

Academic Editor

PLOS ONE